# Evaluating the Effect of Framing Energy Consumption in Terms of Losses versus Gains on Air-Conditioner Use: A Field Experiment in a Student Dormitory in Japan

**Shimpei Iwasaki [1],\*, Samuel Franssens [2], Siegfried Dewitte [3] and Florian Lange [3]**

[1]  Department of Environmental Science, Fukuoka Women's University, Fukuoka 813-8529, Japan
[2]  Marketing Department, Rennes School of Business, 35065 Rennes, France; samuel.franssens@rennes-sb.com
[3]  Behavioral Economics and Engineering Group, KU Leuven, 3000 Leuven, Belgium; Siegfried.dewitte@kuleuven.be (S.D.); florian.lange@kuleuven.be (F.L.)
\*  Correspondence: iwasaki@fwu.ac.jp; Tel.: +81-9269-23115

**Abstract:** Promoting energy conservation in university dormitories is challenging because student residents are typically charged a flat utility fee. One possibility to curb excessive energy use in the absence of monetary incentives is to highlight the environmental consequences of energy use. However, it is still largely unknown how these consequences should be communicated to effectively change people's behavior. In the present study, we analyzed the effect of framing the environmental consequences of energy use in terms of losses versus gains on the air-conditioner use of student residents of a Japanese university dormitory. A total of 330 students were provided with stickers to attach to the air-conditioner remote control in their dormitory room during the winter term. The stickers conveyed that increasing the temperature will hurt the environment (loss frame), that reducing the temperature will protect the environment (gain frame), or that changing the temperature will affect the environment (neutral frame). Day-to-day variations in objective air-conditioner use data were analyzed as a function of experimental condition to examine the effect of message framing. The change in air-conditioner use from pre-intervention to intervention period did not differ between experimental groups and neither did the change from pre-intervention period to a period after the intervention.

**Keywords:** energy use; pro-environmental behavior; message framing; loss aversion; dormitories; Japan

## 1. Introduction

Rapid and far-reaching transitions in energy use are necessary to mitigate climate change [1]. This requires a thorough understanding of how people can be motivated to moderate their energy use behavior. Promoting energy conservation is a particular challenge in residential halls and university dormitories because student residents are typically charged a flat utility fee. As a consequence, student residents (like hotel guests or office workers) do not have financial incentives to save energy [2–6]. Several attempts have been made to address this lack of monetary incentives by linking energy use to social or material rewards [2–9]. A possible limitation of this approach is that the effects of such rewards may be relatively short-lived and that energy use may return to baseline once the intervention is discontinued ([10–12], but see [13]). Here, we adopt an alternative approach by informing people about a potentially valued natural consequence of their energy use: its impact on the environment.

Previous evaluations of environmental impact messages (often also referred to as biospheric appeals) have found mixed results. Some studies have found positive effects on conservation behavior or its proposed psychological antecedents [14–17], whereas others found no such effects [18–20].

Critically, the effectiveness of environmental impact messages may depend on how the consequences of people's behavior are framed [21]. For example, messages may be

more effective when the consequences of environmentally relevant behaviors are framed in terms of losses rather than gains [22–24]. In other words, attempts to promote energy conservation may be more successful when they convey that energy-intensive behavior will hurt the environment rather than that energy-saving behavior will help the environment. Such superiority of loss versus gain framing may be due to people finding the current state of the environment satisfactory (i.e., good enough), so that the prevention of negative change would be more motivating than the promotion of positive change. However, these findings need to be taken with a grain of salt as they are largely based on self-report studies examining attitudes and intentions rather than actual energy use behavior (see [25], for a discussion of the validity issues related to such self-report proxies). In fact, a recent review of 61 framing studies in the environmental domain [24] included only seven studies on actual behavior, only one of which [26] included a measure of energy use. Moreover, the one study that examined the effect of loss framing on actual energy use behavior [26] confounded the framing intervention with additional interventions, which makes it impossible to attribute the intervention effect to the loss framing. As a result, it is still largely unclear whether framing the environmental impact of energy use behavior in terms of losses versus gains can promote actual energy conservation.

In the present study, we analyzed the effect of differentially framed environmental impact messages on air-conditioner use of student residents of a Japanese university dormitory. We provided residents with stickers to attach to the air-conditioner remote control in their dormitory room. These stickers highlighted the impact of air-conditioner use in terms of losses, gains, or neutral effects. By comparing energy use before, during, and after this intervention, we examined which of the three message frames was most effective in reducing air-conditioner use.

## 2. Methodology

### 2.1. Study Site

The study took place in a female student dormitory at Fukuoka Women's University, Japan. All new local and foreign exchange students who enter the university are required to live in the dormitory for almost one year from April to February. They live together in groups of up to four (three local and one foreign student) in one of 84 units distributed across three buildings. Each unit consists of four private single rooms (each including a balcony) as well as a shared dining kitchen, bathroom, and lavatory. At the time of the study (November 2018 to January 2019), 330 student residents lived in such four-person dormitory units and were included in our study. Dormitory residents pay 6300 Japanese yen per month (approximately USD 60, EUR 50 at the time of the study) as an all-inclusive utility fee that covers electricity, gas, and water use. That is, they do not have a monetary incentive for energy conservation.

Air-conditioners are installed in the private single rooms and the shared dining kitchen, but we focused only on the air-conditioner data from private single rooms. These air-conditioners are used for cooling in the summer and for heating in the winter. Our decision to focus on air-conditioner data was motivated partly by practical concerns and partly by the earlier finding that air-conditioner use accounts for the largest share of electricity use in the dormitory [27].

### 2.2. Procedure

We created three types of environmental impact messages (loss frame, gain frame, and neutral frame) to communicate the environmental consequences of air-conditioner use (see Figure 1). The loss-frame sticker indicated that choosing higher temperatures (i.e., temperatures that deviate more from the low outside temperatures during the time of the study) would hurt the environment. It used a color gradient from grey to red, with the red pole (i.e., 24 °C) denoting the level that would hurt the environment the most. The gain-frame sticker indicated that choosing lower temperatures (i.e., temperatures that deviate less from outside temperatures) would protect the environment. It used a color

gradient from green to grey, with the green pole (i.e., 18 °C) denoting the level that would protect the environment the most. The neutral frame sticker depicted an exclusively grey color scale and did not use positively or negatively valenced terms to describe the effect of temperature settings on the environment. We used English messages on the stickers because the dormitory residents included foreign students and the Japanese students were expected to understand English.

**Figure 1.** The stickers used in the experiment. Left: loss-frame condition, middle: gain-frame condition, right: neutral frame condition.

The dormitory students living in the unit buildings described above were divided into the three experimental groups (loss frame: *n* = 111, gain frame: *n* = 110, neutral frame: *n* = 109). Each member of a unit (consisting of three or four students) received the same sticker. The stickers were distributed on 3 December 2018 by means of letters sent to the leaders of each unit. This letter included information about the general aims of the study and instructions for attaching the sticker to the air-conditioner remote control in students' private rooms (see Figure 2). Unit leaders were asked to pass on these instructions and the stickers to the other students living in the unit. On 21 December 2018, students received a text message on messaging application LINE requesting them to remove the sticker before the end of the winter break (7 January 2019).

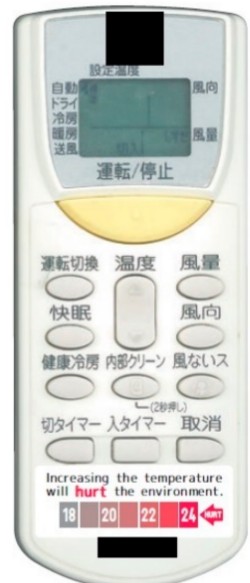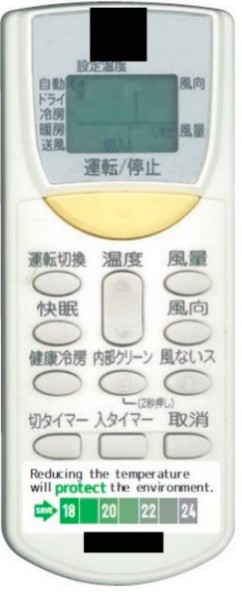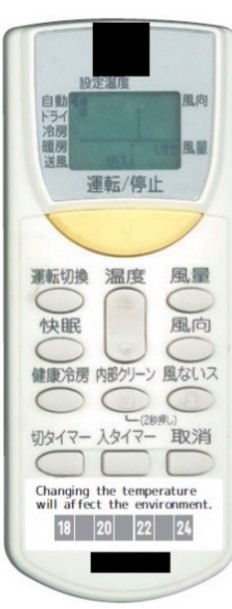

**Figure 2.** The stickers applied on the air-conditioner remote control.

### 2.3. Data and Analysis

From the administrative office of Fukuoka Women's University, we obtained data on the energy used by the air conditioners in 330 rooms, grouped into 84 units, on 92 days, beginning on Thursday 11 January 2018 and ending on Thursday 31 January 2019. As each room within a unit was in the same experimental condition (28 units in each condition) and because air conditioner use within units is not independent, we analyzed the data at the level of the unit. The unit of analysis is the average energy used by an air conditioner within a unit, measured in kilowatt-hours (we averaged because the number of occupied

rooms was different across the units: 78 units had 4 occupied rooms and 6 had 3 occupied rooms). This results in a fully balanced data set of 7728 observations.

The intervention started on 4 December 2018 and ended on 21 January 2018. To test whether the change in energy consumption from pre-intervention to intervention and post-intervention period was smaller in the loss or gain frame conditions than in the control condition, we conducted a panel analysis with *condition* (*loss frame* vs. *gain frame* vs. *neutral frame*) and *period* (*post* is one day after the intervention or later, *pre* is one day before the intervention or earlier, *during* otherwise) as independent variables.

In the statistical model, we included some fixed effects to control for differences between units and time periods that may affect air-conditioner use. *Building*, *floor*, and the interaction between *building* and *floor* controlled for differences in the spatial orientation of the rooms. *Weekday* (Monday, Tuesday, etc.) and a term named *holiday*, indicating whether the students had class on a particular day, controlled for differences between the days. We obtained meteorological information for Fukuoka for the days of the experiment [28]: *temperature* measured in Celsius, *air pressure* measured in millibars, *humidity* percentage, and *wind speed* measured in kilometers per hour.

The data were not normally distributed because there were many days with no air-conditioner use. Therefore, we took the double logarithm of the daily energy use, which rendered the residuals more normal. Together with the independent variables, we got the following model:

$$
\begin{aligned}
log(log(Y_{it}+1)+1) \\
= building_i + floor_i + building_i \times floor_i + weekday_t \\
+ holiday_t + temperature_t + airpressure_t + humidity_t \\
+ windspeed_t + condition_i + period_t \\
+ condition_i \times period_t + \varepsilon_i
\end{aligned} \tag{1}
$$

where *i* refers to unit and *t* refers to period.

## 3. Results

Figure 3 shows the daily energy use, averaged per condition, across time. Figure 4 shows a violin plot that depicts the distribution of the average energy use per unit, per period. Figure A1 shows the residuals of the model with only the covariates. As the figures suggest, there was no difference between the loss frame condition and the neutral condition in the increase from pre-intervention to intervention period (in the following, all estimates are on the double log scale: *estimate* = 0.003, $t(7627) = 0.26$, $p = 0.791$) or the increase from pre-intervention to post-intervention period (*estimate* = 0.009, $t(7627) = 0.92$, $p = 0.358$). The same holds for the comparison between the gain frame condition and the neutral condition: no difference in the increase from pre-intervention to intervention period (*estimate* = 0.003, $t(7627) = 0.2$, $p = 0.839$) or the increase from pre-intervention to post-intervention period (*estimate* = 0.011, $t(7627) = 1.1$, $p = 0.271$). Finally, also when comparing the loss frame condition and the gain frame condition, there was no difference in the increase from pre-intervention to intervention period (*estimate* = $-0.001$, $t(7627) = -0.06$, $p = 0.951$) or the increase from pre-intervention to post-intervention period (*estimate* = 0.002, $t(7627) = 0.18$, $p = 0.855$). Appendix A reports the full regression table. The interactions between condition and period remained nonsignificant when we tested them without controlling for covariates. The results do not change if we drop the assumption that air conditioner use within units is related and analyze the data at the level of the individual air conditioner.

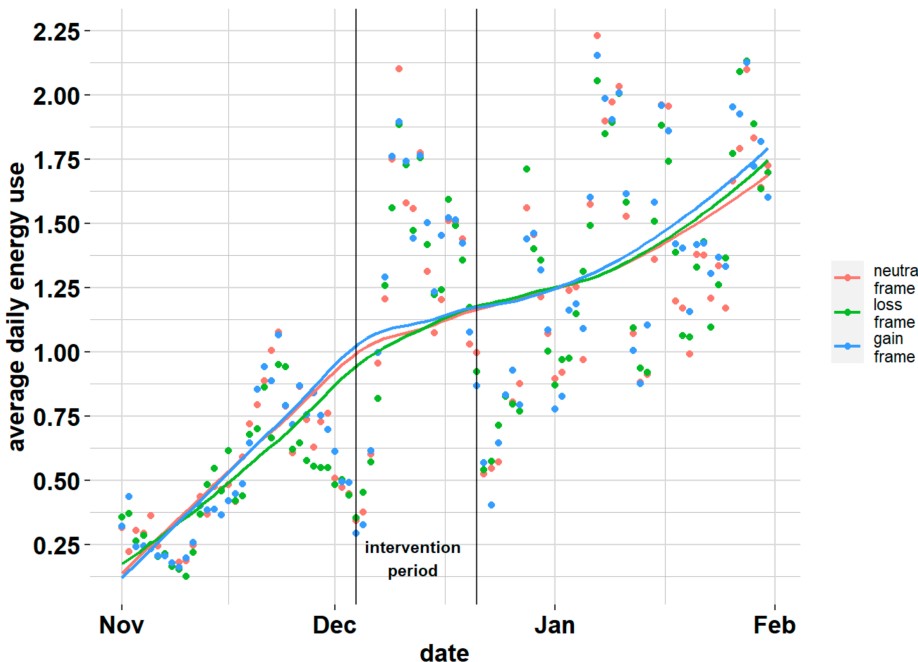

**Figure 3.** Average daily energy use per experimental condition in kilowatt-hours.

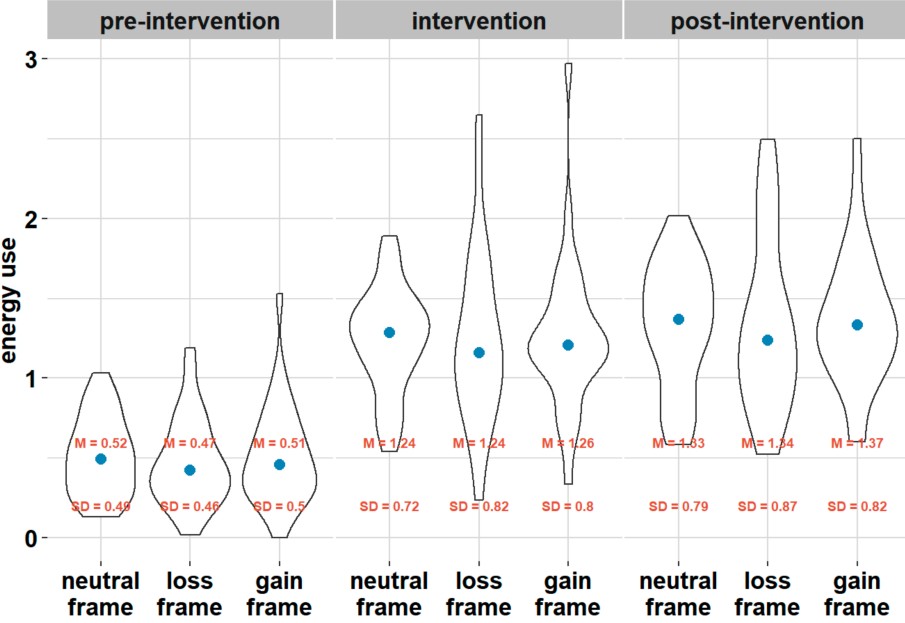

**Figure 4.** Violin plot of the average energy use per unit, per period. Blue dots represent medians.

## 4. Discussion

Using objective energy use data with high temporal resolution, the present study investigated the effect of differentially framed environmental impact messages on the air-conditioner use of student residents of a Japanese university dormitory. We did not find these messages to be more effective when students' impact on the environment was framed in terms of losses rather than gains. Moreover, the energy use of students receiving loss-framed or gain-framed messages did not differ from energy use of students in a neutrally framed control condition.

In line with several recent studies (e.g., [18–20]), the present study did not find evidence for the effect of environmental impact messages on environmentally relevant behavior. However, it must be noted that each of our three experimental conditions (even

the neutral control condition) involved some kind of information about the environmental consequences of participants' air-conditioner use. This information was arguably more vague in the control condition, but the differential effect between the conditions might have been too small to be detectable in our study. The same applies to the critical comparison between the loss-frame condition and the gain-frame condition. We took several measures to ensure that the stickers used in these conditions were as similar as possible on all dimensions except the framing of the message. Because of that, the difference between the two conditions may have been too subtle to exert significant effects in a field experiment. In comparison to laboratory studies, field experiments on environmentally relevant behavior are often more noisy and less controlled [29]. This high level of noise might critically attenuate the effect of behavioral interventions. For example, in the present study, we were not able to ensure that every participant was similarly exposed to the experimental manipulation, because we could not check whether they attached the sticker according to our instructions and paid attention to the sticker when making temperature setting decisions in everyday life. Future field experiments testing the effect of gain vs. loss frame messages on energy use may benefit from implementing less subtle interventions, for example, highly attention-grabbing posters pre-installed in the shared parts of dormitories. In addition, a larger sample size will likely be necessary to detect the effects of such interventions. Pooling data from multiple study sites (e.g., through using the Psychological Science Accelerator [30]) may be one strategy to meet these sample-size demands. Such multisite studies may also increase the heterogeneity of the recruited samples. The present study exclusively included female students studying at the same Japanese institution of higher education. Further research in different populations is necessary to examine the generality of message framing effects on energy use.

**Author Contributions:** Conceptualization, S.I., S.F., S.D. and F.L.; methodology, S.I., S.F., S.D. and F.L.; data curation, S.I.; data analysis, S.F. and S.I.; writing—draft, S.I., S.F. and F.L.; writing—review and editing, S.I., S.F., S.D. and F.L. All authors have read and agreed to the published version of the manuscript.

**Funding:** This research received no external funding.

**Institutional Review Board Statement:** The study was approved by the Fukuoka Women's University Epidemiology and Ethics, etc. Research Review Committee (No. 2018-26).

**Informed Consent Statement:** Informed consent was obtained from all subjects involved in this study using an opt out approach.

**Data Availability Statement:** The data presented in this study are available on request from the corresponding author.

**Acknowledgments:** The authors thank Mina Nakagawa for supporting the field intervention. Florian Lange was supported by a FWO postdoctoral fellowship (No 12U1221N).

**Conflicts of Interest:** The authors declare no conflict of interest. The funders had no role in the design of the study; in the collection, analyses, or interpretation of data; in the writing of the manuscript, or in the decision to publish the results.

## Appendix A

**Table A1.** Full regression table.

| Term | Estimate | Standard_Error | T | Df | P |
|---|---|---|---|---|---|
| (Intercept) | 0.4234 | 0.0429 | 9.879 | 68 | 0.0000 |
| temperature | −0.0441 | 0.0010 | −43.639 | 7627 | 0.0000 |
| airpressure | −0.0064 | 0.0007 | −8.804 | 7627 | 0.0000 |
| humidity | −0.0001 | 0.0003 | −0.244 | 7627 | 0.8069 |

**Table A1.** *Cont.*

| Term | Estimate | Standard_Error | T | Df | P |
|---|---|---|---|---|---|
| windspeed | −0.0133 | 0.0020 | −6.559 | 7627 | 0.0000 |
| weekdayThursday | −0.0073 | 0.0068 | −1.074 | 7627 | 0.2827 |
| weekdayFriday | −0.0169 | 0.0070 | −2.420 | 7627 | 0.0155 |
| weekdaySaturday | 0.0605 | 0.0083 | 7.293 | 7627 | 0.0000 |
| weekdaySunday | 0.0714 | 0.0083 | 8.564 | 7627 | 0.0000 |
| weekdayTuesday | 0.0028 | 0.0069 | 0.405 | 7627 | 0.6854 |
| weekdayWednesday | −0.0075 | 0.0070 | −1.074 | 7627 | 0.2826 |
| holidaylecture_day | −0.1539 | 0.0062 | −24.716 | 7627 | 0.0000 |
| periodintervention | 0.0778 | 0.0098 | 7.965 | 7627 | 0.0000 |
| periodpost_intervention | 0.0526 | 0.0094 | 5.596 | 7627 | 0.0000 |
| buildingB | −0.0781 | 0.0607 | −1.286 | 68 | 0.2027 |
| buildingC | 0.0725 | 0.0555 | 1.306 | 68 | 0.1961 |
| floor3 | −0.0101 | 0.0555 | −0.183 | 68 | 0.8557 |
| floor4 | 0.0409 | 0.0555 | 0.737 | 68 | 0.4636 |
| floor5 | 0.0316 | 0.0554 | 0.571 | 68 | 0.5699 |
| floor6 | 0.0540 | 0.0555 | 0.972 | 68 | 0.3343 |
| conditiongain_frame | 0.0087 | 0.0279 | 0.311 | 68 | 0.7564 |
| conditionloss_frame | −0.0154 | 0.0281 | −0.546 | 68 | 0.5870 |
| conditiongain_frame:periodintervention | 0.0025 | 0.0125 | 0.203 | 7627 | 0.8392 |
| conditiongain_frame:periodpost_intervention | 0.0110 | 0.0100 | 1.102 | 7627 | 0.2705 |
| conditionloss_frame:periodintervention | 0.0033 | 0.0125 | 0.264 | 7627 | 0.7915 |
| conditionloss_frame:periodpost_intervention | 0.0092 | 0.0100 | 0.919 | 7627 | 0.3581 |
| buildingB:floor3 | 0.1481 | 0.0858 | 1.726 | 68 | 0.0889 |
| buildingB:floor4 | 0.0289 | 0.0862 | 0.335 | 68 | 0.7388 |
| buildingB:floor5 | 0.0485 | 0.0858 | 0.566 | 68 | 0.5735 |
| buildingB:floor6 | 0.1858 | 0.0890 | 2.089 | 68 | 0.0405 |
| buildingC:floor3 | −0.0493 | 0.0784 | −0.629 | 68 | 0.5315 |
| buildingC:floor4 | −0.0035 | 0.0802 | −0.043 | 68 | 0.9658 |
| buildingC:floor5 | −0.0365 | 0.0822 | −0.444 | 68 | 0.6584 |

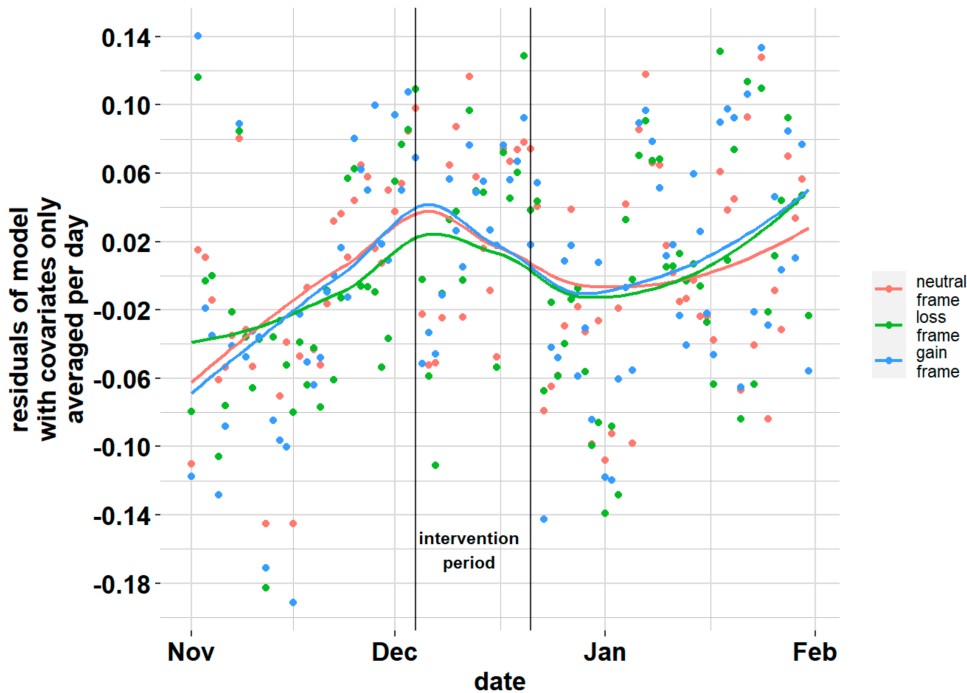

**Figure A1.** Residuals, averaged per day, of the model with only covariates, per condition.

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
