# Peer review of "Evaluating the Effect of Framing Energy Consumption in Terms of Losses versus Gains on Air-Conditioner Use: A Field Experiment in a Student Dormitory in Japan"

_sustainability, doi:10.3390/su13084380_

Round 1
Author Response
Thank you very much for reviewing our manuscript.
The manuscript has been carefully re-checked, and the appropriate changes have been made.
The responses to the reviewers' comments are attached herewith.
We thank the reviewers for their thoughtful suggestions and insights, which have improved the manuscript and produced a more balanced and better account of the research. We hope that the revised manuscript is now suitable for publication in Sustainability.

Reviewer 2 Report
Thank you to the authors for this interesting work. The paper is in good shape and I recommend some points for clarification/ improvement:
- the stimuli seem intuitive and well-thought, but were these piloted?
- this is a cluster randomization. Although you measured individual data, the randomization unit was the apartment (because all students received the same). Please consider needed adjustment when reporting samples sizes per condition and data analysis accounting for cluster design.
- Consider elaborating a bit more on the conceptual part. There is mixed evidence but are there good reasons for a hypothesis in favor of loss-frame? Look into the health literature and possible similarities/ differences
- The same applies to the Discussion. Elaborate on possible conceptual reasons (not just methodological) that may have accounted for the lack of effect.
Best wishes
Author Response
Thank you very much for reviewing our manuscript and making some comments on it. The manuscript has been carefully re-checked, and the appropriate changes have been made in accordance with the reviewers' suggestions. The responses to the reviewers' comments are attached herewith.
We thank the reviewers for their thoughtful suggestions and insights, which have improved the manuscript and produced a more balanced and better account of the research. We hope that the revised manuscript is now suitable for publication in Sustainability.
